# Risk factors, predictions, and progression of acute kidney injury in hospitalized COVID-19 patients: An observational retrospective cohort study

**Maryam N. Naser** *, Rana Al-Ghatam, Abdulla H. Darwish, Manaf M. Alqahtani, Hajar A. Alahmadi[☯], Khalifa A. Mohamed[☯], Nahed K. Hasan, Nuria S. Perez**

Bahrain Defence Force Hospital, Royal Medical Services, Riffa, Kingdom of Bahrain

☯ These authors contributed equally to this work.
* mariam.naji@bdfmedical.org, dr_marriam_naji@yahoo.com

**Data Availability Statement:** All relevant data are within the manuscript and its Supporting Information files.

## Abstract

### Objectives

Studies have shown that acute kidney injury (AKI) occurrence post SARS-CoV-2 infection is complex and has a poor prognosis. Therefore, more studies are needed to understand the rate and the predications of AKI involvement among hospitalized COVID-19 patients and AKI's impact on prognosis while under different types of medications.

### Patients and methods

This study is a retrospective observational cohort study conducted at Bahrain Defence Force (BDF) Royal Medical Services. Medical records of COVID-19 patients admitted to BDF hospital, treated, and followed up from April 2020 to October 2020 were retrieved. Data were analyzed using univariate and multivariate logistic regression with covariate adjustment, and the odds ratio (OR) and 95% confidence (95% CI) interval were reported.

### Results

Among 353 patients admitted with COVID-19, 47.6% developed AKI. Overall, 51.8% of patients with AKI died compared to 2.2% of patients who did not develop AKI (p< 0.001 with OR 48.6 and 95% CI 17.2–136.9). Besides, deaths in patients classified with AKI staging were positively correlated and multivariate regression analysis revealed that moderate to severe hypoalbuminemia (<32 g/L) was independently correlated to death in AKI patients with an OR of 10.99 (CI 95% 4.1–29.3, p<0.001). In addition, 78.2% of the dead patients were on mechanical ventilation. Besides age as a predictor of AKI development, diabetes and hypertension were the major risk factors of AKI development (OR 2.04, p<0.01, and 0.05 for diabetes and hypertension, respectively). Also, two or more comorbidities substantially increased the risk of AKI development in COVID-19 patients. Furthermore, high levels upon hospital admission of D-Dimer, Troponin I, and ProBNP and low serum albumin were associated with AKI development. Lastly, patients taking ACEI/ARBs had less chance to develop AKI stage II/III with OR of 0.19–0.27 (p<0.05–0.01).

**Funding:** The author(s) received no specific funding for this work.

**Competing interests:** The authors have declared that no competing interests exist.

## Conclusions

The incidence of AKI in hospitalized COVID-19 patients and the mortality rate among AKI patients were high and correlated with AKI staging. Furthermore, laboratory testing for serum albumin, hypercoagulability and cardiac injury markers maybe indicative for AKI development. Therefore, clinicians should be mandated to perform such tests on admission and follow-up in hospitalized patients.

## Introduction

Since December 2019, a novel coronavirus causing the severe acute respiratory syndrome called SARS-CoV2 has caused fear in the whole world. The main reasons for this fear are two-folds; its fast-spreading infection and clinical course is variable and unpredictable, ranging from asymptomatic infection to multi-organ system failure and death [1–3].

Initially, a study consisting of 116 patients from Wuhan in early 2020 reported that none of 116 hospitalized COVID-19 patients developed acute kidney injury (AKI), suggesting that AKI is uncommon following SARS-CoV2 infection [3]. However, several studies confirmed that AKI occurred in 8–17% of hospitalized COVID-19 patients [4], and may increase to 20–40%, mostly in critically ill patients with COVID-19 [5–7]. Furthermore, it has been recognized that developing AKI is a poor prognostic factor in COVID-19 infection [4, 8].

The reasons for developing AKI in COVID-19 patients are most likely multiple direct and indirect dependent pathways. SARS-CoV-2 infects lung epithelial cells via binding to angiotensin-converting enzyme II (ACE2) receptor expressed on these cells [9]. However, ACE2 RNA was found to be expressed in gastrointestinal organs (small intestine, duodenum) and kidneys [10–12]. Besides, SARS-CoV-2 mRNA was found in 15% of patient's plasma indicating viremia [2]. Furthermore, autopsies have shown that viral particles were present in renal endothelial cells and endothelial injury in the kidneys and the lungs, suggesting a direct action causing AKI [13]. Secondly, SARS-CoV-2 causes pneumonia, and patients may develop right and/or left ventricular failures, which will lead to kidney congestion or kidney hypoperfusion, respectively [2, 14]. Thirdly, a dysregulated immune response following infection might occur, resulting in lymphopenia and cytokine syndrome [15–17]. Fourthly, SARS-CoV-2 infection may induce endothelialitis and hypercoagulability resulting in microthrombi or microembolism that cause kidney infarction [16, 18]. Therefore, the complexity of AKI occurrence post SARS-CoV-2 infection is evident, and further investigations are warranted to explain the predictions, risk factors, and possible prevention of AKI involvement.

Based on the above and our clinical observations, we sought to study the rate of AKI involvement among patients hospitalized with COVID-19 infection and AKI's impact on prognosis, morbidity, and mortality in Bahrain. Furthermore, a detailed analysis of laboratory testing at the time of admission and the prediction to develop AKI was determined. Lastly, which medication might have an influence on AKI prevention/progression in COVID-19 patients were investigated.

## Patients and methods

### Study protocol and design

A study protocol was submitted and approved by the Research & Research Ethics Committee at the Bahrain Defence Force (BDF), Royal Medical Services (#BDF/R&REC/2020-462). Also, this research study was approved by the Bahrain National COVID-19 Research Team (CRT-COVID2020-061).

This study is a retrospective observational cohort conducted at the BDF, Royal Medical Services in Bahrain. All COVID-19 patients' data and laboratory results admitted to BDF hospital, treated, and followed up from April 2020 to October 2020 were retrieved and analyzed.

## Patients

Three hundred and fifty-three male patients' data were retrieved and analyzed. The 353 patients' data included in the study were based on the following inclusion criteria of a) age of $\geq 15$ years old, b) positive for SARS-CoV 2 by real-time PCR, and c) all enrolled positive patients admitted into intensive care/isolation unit (ICU) only for COVID-19 patients. The exclusion criteria were a) age $< 14$ years old, b) patients with eGFR $<15$ ml/min (CKD of 5) with or without renal replacement therapy before admission, and c) patients with short hospital stay ($< 48$ hours). Furthermore, none of the patients had a history of immunodeficiency disorders.

## AKI classification and pneumonia severity index

AKI was classified based on KDIGO classification [19]. Accordingly, AKI has been classified as stage 1 with an increase in serum creatinine $\geq 26.5$ μmol/L or $\geq 50\%$ within 48 h, stage 2 with an increase 2–3 times the baseline, or stage 3 with $> 3$ times baseline or at least 354 μmol/L or needs a renal replacement therapy (RRT).

Pneumonia severity index (PSI) was calculated based on demography, comorbidities, physical exam, vital signs, and laboratory and imaging results risk class points. The risk class points were as follows: class I ($<50$), class II (51–70), class III (71–90), class IV (91–130), and class IV ($>130$) [20].

## Data collection

All COVID-19 positive patients went through routine laboratory tests as specified in the Bahrain COVID-19 Protocol, which includes CBC, kidney, and liver function tests, urine analysis, ESR, C-reactive protein (CRP), prothrombin time (PT), ferritin, cardiac enzymes, troponin I/T, ProBNP, and procalcitonin (PCT). All the laboratory, clinical, and demographic data were obtained from electronic medical records using a data collection tabulation.

## Data analysis

We performed a descriptive statistical analysis based on the skewness score of $> 2$. Since most of the laboratory data parameters for the AKI patients were skewed, results were expressed as median (interquartile range, IQR), whereas categorical data were reported as percentages. To test for significance, the non-parametric Mann Whitney was applied between two groups, and the non-parametric Kruskal–Wallis test for medians was used to compare the quantitative variables among more than two groups, when appropriate. To identify risk factors associated with the development of AKI, a multinomial logistic regression model was applied with the adjustment for covariates such as age. The estimated effect was reported by adjusted odds ratios (ORs) with their 95% confidence intervals (95% CIs). A p-value of $<0.05$ was considered to be statistically significant. All analyses were performed using SPSS 25 statistical package.

# Results

## Characteristics of the study population

From April 2020 to October 2020, a total of 353 male COVID-19 patients were admitted to BDF hospital, with a mean age of 55.8 (±15.7) years. All patients' records were followed from admission till the day of discharge or the day of death. Of these, 168 patients (47.6%) developed

**Table 1. Characteristics and medical conditions of the study subjects.**

| Parameter | Patients with no AKI | Patients with AKI | P value |
|---|---|---|---|
| Age (overall) | 49 ± 23 | 65 ± 15 | <0.001 |
| Age (Bahraini's) | 53.7± 15.9 | 65.8 ± 11.7 | <0.001 |
| Age (non-Bahraini's) | 42.6 ± 11.2 | 52.2 ± 10.1 | <0.001 |
| | *Percentages* | | |
| Nationality (Bahraini) | 56.2 | 82.1 | <0.001 |
| Comorbidity | 55.7 | 86.9 | <0.001 |
| Diabetes | 30.3 | 66.1 | <0.001 |
| Hypertension | 26.5 | 66.1 | <0.001 |
| Chronic Kidney Disease | 0 | 25 | <0.001 |
| Cardiovascular Disease | 5.9 | 24.4 | <0.001 |
| Chronic Pulmonary Disease | 7 | 7.4 | n.s. |
| Chronic Hepatic Disease | 0 | 1.2 | n.s. |
| Cough | 62.2 | 63.1 | n.s. |
| Fever | 68.6 | 61.9 | n.s. |
| Dyspnea | 45.4 | 55.3 | n.s. |
| Diarrhea | 13.5 | 10.7 | n.s. |
| URT | 29.7 | 20.2 | <0.05 |
| Fatigue | 45.4 | 36.9 | n.s. |
| Hypoxic Encephalopathy | 1.6 | 6 | n.s. |
| Death | 2.2 | 51.8 | <0.001 |

AKI. The age between the AKI and non-AKI groups was significantly different (p<0.001) (Table 1). Furthermore, the frequencies of the comorbidity, diabetes, hypertension, chronic kidney disease (CKD), cardiovascular disease were all significantly higher in the patients with AKI (p<0.001). However, there was no significant difference in the frequency of patients with chronic pulmonary and hepatic diseases and hypoxic encephalopathy. Besides, COVID-19 symptoms such as cough, fever, dyspnea, diarrhea, or fatigue were not significantly different between AKI and non-AKI groups.

## COVID-19 patients who developed AKI had higher levels of inflammatory, cardiac injury, and fibrinolysis markers at time of admission

On admission, several laboratory tests were performed on all COVID-19 positive patients. Upon analysis, CRP, ESR, ferritin, and PCT inflammatory markers were significantly higher in patients that developed AKI than in patients that did not develop AKI (Table 2). Furthermore, cardiac injury markers consisting of CK, CK-MB, Troponin T levels were significantly higher in patients that developed AKI than in non-AKI patients. Besides, patients that developed AKI had higher leukocytes and platelets count (p<0.001) but lower lymphocyte count, hemoglobin, and serum albumin levels (p<0.001).

Furthermore, following categorizing AKI stages, laboratory test data at the time of admission are presented in Table 3. Surprisingly, eGFR was significantly lower for patients who developed AKI stage I than stage II or III, and serum creatinine and uric acid values were higher in patients who developed AKI stage I than stage II or III. Besides, leukocytes and CK levels were significantly higher in patients that developed AKI stage II and III.

During the hospital stay, serum creatinine was followed in these patients. Surprisingly, serum creatinine of patients who developed AKI stage I had higher creatinine value on admission (Table 3) and started rising earlier (median day 1) than patients who developed AKI II and III

**Table 2. Laboratory data at the time of admission for the two groups of COVID-19 infected patients.**

| Parameter | Patients with no AKI | Patients with AKI | P value |
|---|---|---|---|
| | Median (IQR) | Median (IQR) | |
| *Kidney function* | | | |
| eGFR (ml/min) | 60 (0) | 57 (24) | <0.001 |
| Creatinine (μmol/L) | 68 (81) | 117 (88) | <0.001 |
| Urea (mmol/L) | 4.6 (2.1) | 9.2 (7.5) | <0.001 |
| Uric Acid (μmol/L) | 281 (121) | 383 (206) | <0.001 |
| Na+ (mmol/L) | 136 (5) | 135 (7) | <0.05 |
| K+ (mmol/L) | 4.4 (0.6) | 4.6 (0.9) | <0.05 |
| HCO3- (mmol/L) | 23 (4) | 21 (5) | <0.001 |
| *Erythropoiesis and Leukopoiesis* | | | |
| Hb (g/dl) | 14.1 (2.6) | 12.9 (3.3) | <0.001 |
| Leukocytes (x$10^3$/μl) | 5.44 (3.09) | 6.47 (5.16) | <0.001 |
| Lymphocytes (x$10^3$/μl) | 1166 (980) | 948 (749) | <0.001 |
| *Clotting and fibrinolysis* | | | |
| Platelets (x$10^3$/μL) | 218 (108) | 201(95) | <0.05 |
| PT (sec) | 14.9 (1.6) | 14.8 (1.8) | n.s. |
| INR | 1.00 (0.12) | 1.05 (0.14) | <0.01 |
| D. Dimer | 0.48 (1.00) | 1.04 (2.00) | <0.001 |
| *Cardiac markers* | | | |
| CK (IU/L) | 152 (340) | 244 (413) | <0.01 |
| CKMB (IU/L) | 27 (20) | 30 (22) | <0.05 |
| Troponin T (μg/L) | 0.00 (0.00) | 0.00 (0.12) | <0.001 |
| ProBNP (pg/ml) | 41 (137) | 304 (940) | <0.001 |
| *Inflammatory markers* | | | |
| ESR (mm/h) | 37 (34) | 50 (35) | <0.001 |
| CRP (mg/L) | 47 (105) | 115 (132) | <0.001 |
| Ferritin (ng/ml) | 704 (1072) | 940 (1234) | <0.01 |
| PCT (ng/ml) | 0.11 (0.10) | 0.4 (1.12) | <0.001 |
| *Liver function* | | | |
| Albumin (g/L) | 39 (7) | 35 (6) | <0.001 |
| ALT (IU/L) | 34 (38) | 29 (26) | <0.01 |
| AST (IU/L) | 41 (31) | 44 (43) | n.s. |
| GGT (IU/L) | 54 (72) | 51 (67) | n.s. |
| ALP (IU/L) | 70 (33) | 76 (39) | n.s. |

(median day 5 and 4.5, respectively, p<0.001) (Fig 1A). However, serum creatinine peak level was significantly less (p<0.01) in AKI stage I patients than AKI stage II and III (Fig 1B). On the other hand, serum albumin at time of admission was not significantly different between patients who developed different stages of AKI (Table 3) but became significantly lower in patients who developed AKI II and III than patients who developed AKI I (p<0.001) (Fig 1C). These low serum albumin values were significantly correlated to proteinuria (p<0.01).

## Univariate and multivariate of comorbidities and laboratory data upon admission associated with AKI development

In our sample, the percent of patients who developed AKI is 47.6%, and the potential predictors of AKI in COVID-19 are summarized in Table 4. Age was a potential predictor for AKI

**Table 3. Laboratory data at the time of admission of the COVID-19 patients that developed different AKI stages.**

| Parameter | Results | | | |
|---|---|---|---|---|
| | AKI Stage I | AKI Stage II | AKI Stage III | P value |
| | Median (IQR) | Median (IQR) | Median (IQR) | |
| *Kidney function* | | | | |
| eGFR (ml/min) | 50.5 (24) | 60 (22) | 60 (24) | <0.05 |
| Creatinine (μmol/L) | 126 (61) | 89 (96) | 91 (99) | <0.01 |
| Urea (mmol/L) | 10.0 (5.9) | 7.1 (11.2) | 8.1 (8.3) | n.s. |
| Uric Acid (μmol/L) | 414 (160) | 359 (188) | 338 (187) | <0.05 |
| Na+ (mmol/L) | 136 (6) | 134 (6) | 136 (5) | n.s. |
| K+ (mmol/L) | 4.60 (0.95) | 4.52 (1.19) | 4.6 (0.8) | n.s. |
| HCO3- (mmol/L) | 21.9 (3.8) | 18.4 (5.2) | 20.9 (4.5) | <0.01 |
| *Erythropoiesis and Leukopoiesis* | | | | |
| Hb (g/dl) | 12.9 (3.0) | 13.5 (3.6) | 12.5 (3.5) | n.s. |
| Leukocytes (x10³/μl) | 5.83 (4.14) | 7.11 (6.71) | 8.11 (5.97) | <0.001 |
| Lymphocytes (x10³/μl) | 1000 (709) | 1044 (1082) | 785 (698) | n.s. |
| *Clotting and fibrinolysis* | | | | |
| Platelets (x10³/μL) | 179 (87) | 200 (103) | 208 (118) | n.s. |
| PT (sec) | 14.8 (1.8) | 14.9 (1.7) | 14.8 (1.8) | n.s. |
| INR | 1.05 (0.13) | 1.05 (0.16) | 1.06 (0.16) | n.s. |
| D. Dimer | 0.92 (1.00) | 0.9 (3.00) | 1.52 (2.00) | n.s. |
| *Cardiac markers* | | | | |
| CK (IU/L) | 196 (351) | 333 (564) | 307 (492) | <0.05 |
| CKMB (IU/L) | 28 (22) | 33 (29) | 31 (21) | n.s |
| Troponin T (μg/L) | 0.00 (0.10) | 0.0 (0.33) | 0.0 (0.15) | n.s. |
| ProBNP (pg/ml) | 299 (752) | 524 (1211) | 258 (971) | n.s. |
| *Inflammatory markers* | | | | |
| ESR (mm/h) | 54 (35) | 48 (33) | 45 (39) | n.s. |
| CRP (mg/L) | 110 (134) | 111 (171) | 125 (165) | n.s. |
| Ferritin (ng/ml) | 785 (1079) | 998 (1638) | 1077 (1795) | n.s. |
| PCT (ng/ml) | 0.31 (0.99) | 0.41 (1.04) | 0.56 (1.76) | n.s. |
| *Liver function* | | | | |
| Albumin (g/L) | 35 (7) | 38 (5) | 34 (6) | n.s. |
| ALT (IU/L) | 27 (25) | 29 (46) | 33 (23) | n.s. |
| AST (IU/L) | 41 (27) | 60 (71) | 48 (42) | <0.05 |
| GGT (IU/L) | 51 (63) | 74 (111) | 48 (70) | n.s. |
| ALP (IU/L) | 77 (35) | 79 (39) | 73 (47) | n.s. |

development. With age adjustment, it was evident that diabetes and hypertension were the major risk factors of AKI development (OR 2.04, p<0.01, and 0.05 for diabetes and hypertension, respectively). However, having two or more comorbidities caused a substantial risk of AKI development in COVID-19 patients (Table 4).

When we categorized laboratory data upon admission (normal vs. high except for lymphocytes, normal vs. <1000/μl), the number of lymphocytes, albumin, troponin I, CRP, ferritin, D-dimer, and ProBNP, were associated with AKI development (Table 5). However, following age adjustment, only albumin, D-Dimer, Troponin I, and ProBNP high values were associated with AKI development (Table 5).

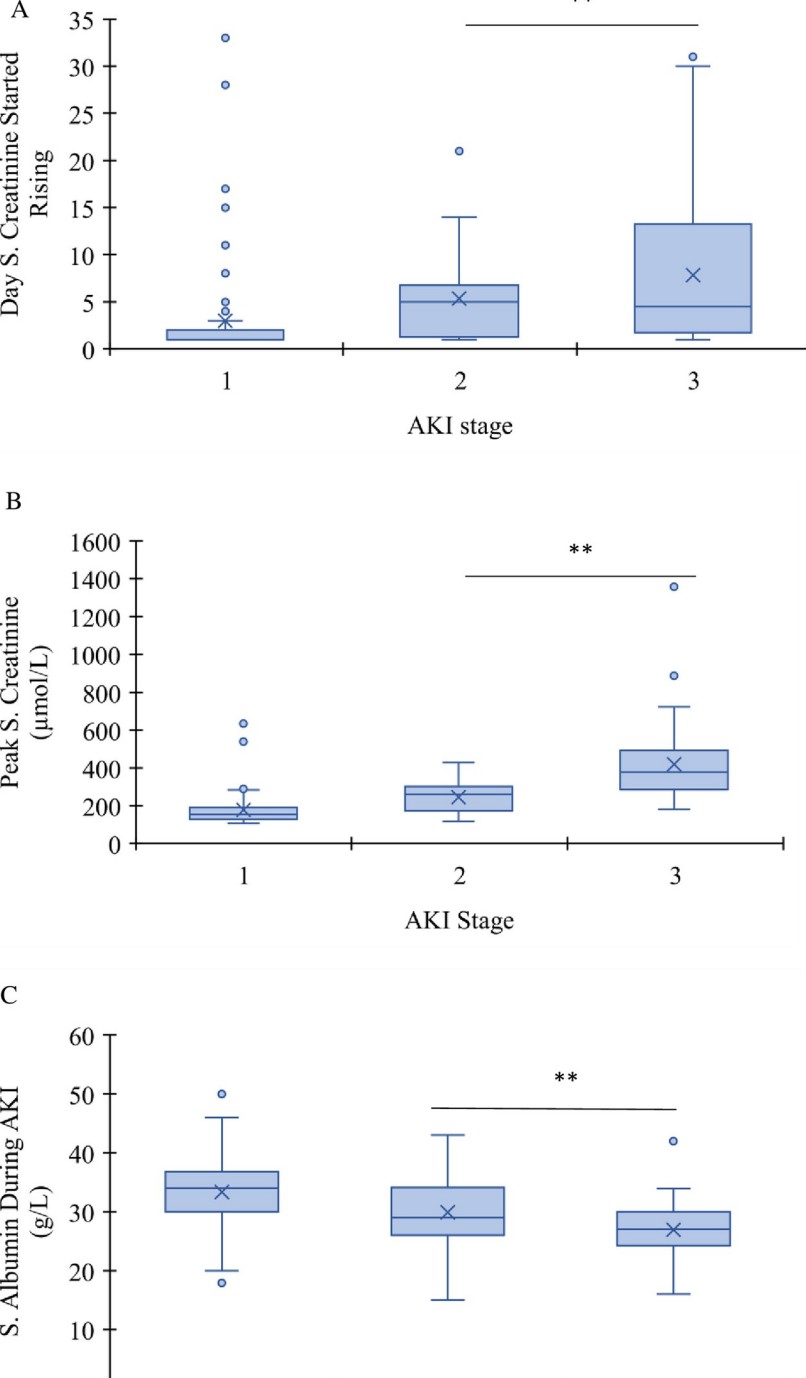

**Fig 1.** Box plot of number of days for serum creatinine started rising following hospitalization (A), peak level of serum creatinine (B), and serum albumin (C) in AKI groups. In either case, AKI stage I was significantly different than AKI stage II or III (For serum creatinine was faster to rise, but less peak value of creatinine, and for serum albumin was higher; ∗∗p<0.001).

**Table 4. Univariate and multivariate logistic regression analysis of comorbidities associated with AKI development.**

| Variable | OR (unadjusted) | 95% CI | P value | OR (Adjusted)* | 95% CI | P value |
|---|---|---|---|---|---|---|
| Age (yr) | 1.05 | 1.03–1.07 | <0.001 | - | - | - |
| Nationality Bahrain/Non-Bahrain | 3.58 | 2.20–5.85 | <0.001 | 1.50 | 0.85–2.65 | n.s. |
| Diabetes | 2.60 | 1.57–4.30 | <0.001 | 2.04 | 1.19–3.50 | <0.01 |
| Hypertension | 3.16 | 1.87–5.35 | <0.001 | 2.04 | 1.16–3.61 | <0.05 |
| Cardiovascular Disease | 2.25 | 1.04–4.89 | <0.05 | 1.73 | 0.76–3.92 | n.s. |
| Chronic Pulmonary Disease | 0.70 | 0.28–1.76 | n.s | 0.53 | 0.20–1.40 | n.s. |
| Any 1 Comorbidity | 2.98 | 1.63–5.44 | <0.001 | 1.90 | 0.99–3.67 | n.s. |
| Any 2 Comorbidity | 5.76 | 2.99–11.10 | <0.001 | 2.56 | 1.24–5.30 | <0.05 |
| Any 3 or more Comorbidity | 14.62 | 7.16–29.82 | <0.001 | 5.70 | 2.59–12.52 | <0.001 |

*Adjusted for age.

## Pneumonia severity in COVID-19 patients without and with AKI

Of the 353 COVID-19 patients, PSI was classified from 1 to 5. Accordingly, it is evident that the severity of pneumonia is related to AKI involvement (Fig 2). In non-AKI, 88.5% of patients had a PSI scale of 1 or 2, whereas 89–100% of AKI patients had a PSI scale of 2–5 (p<0.001). Nevertheless, in AKI patients, PSI was not associated with the percent of patients who went on mechanical ventilation nor died.

## Recovery versus death of COVID-19 patients with AKI development

Overall, patients with AKI had 87 deaths (51.8%) in comparison to 4 deaths (2.2%) in patients that did not develop AKI (p< 0.001 with OR 48.6 and 95% CI 17.2–136.9) (Fig 3A). The Kaplan-Meier survival curve showed that 50% of patients with AKI died within 33 days of admission to the ICU/isolation unit (Fig 3A). It must be mentioned that there were 21 transferred patients, whom 6 out of 7 of them died (~7% of the death within AKI patients) within 2–28 days. This indicates that median survival time of 33 days should be a slightly higher.

The percentage of cumulative recovery rate in COVID-19 patients with AKI is directly related to AKI staging. Approximately 35% of AKI stage 1 patients started to recover after 7 days of admission; this percentage went up to ~50% after 2 weeks and plateaued after that (p<0.0001 vs. stages 2 and 3) (Fig 3B).

**Table 5. Multivariate logistic regression analysis of inflammatory and cardiac markers results upon hospital admission associated with AKI development.**

| Variable | OR (unadjusted) | 95% CI | P value | OR (Adjusted)* | 95% CI | P value |
|---|---|---|---|---|---|---|
| Age (yr) | 1.08 | 1.06–1.10 | <0.001 | - | - | - |
| Lymphocytes (low vs. normal) | 1.79 | 1.17–2.74 | <0.01 | 1.12 | 0.69–1.83 | n.s. |
| Albumin (low vs. normal) | 6.89 | 3.98–11.89 | <0.001 | 4.53 | 2.52–8.15 | <0.001 |
| CRP (high vs. Normal) | 4.22 | 1.88–9.49 | <0.001 | 1.50 | 0.59–3.82 | n.s. |
| Ferritin (high vs. normal) | 2.75 | 1.13–6.69 | <0.05 | 2.77 | 0.93–8.27 | n.s. |
| D Dimer (high vs. normal) | 4.26 | 2.62–6.93 | <0.001 | 2.59 | 1.51–4.42 | <0.001 |
| Troponin (high vs. normal) | 4.55 | 1.66–12.47 | <0.01 | 4.26 | 1.39–13.10 | <0.05 |
| CK-MB | 1.52 | 0.985–2.34 | n.s. | 1.284 | 0.79–2.09 | n.s. |
| ProBNP | 4.05 | 2.23–7.34 | <0.001 | 2.10 | 1.07–4.01 | <0.05 |

*Adjusted for age.

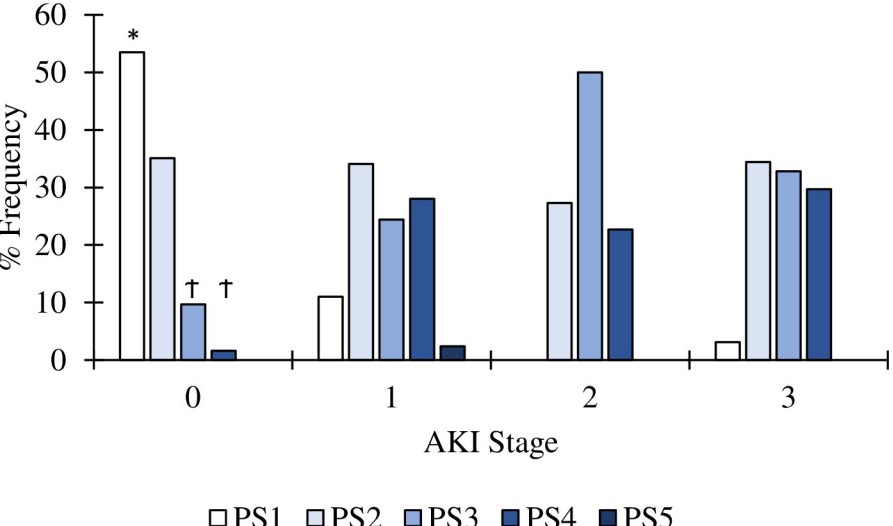

**Fig 2. The frequency of non-AKI and different stages of AKI patients who had pneumonia and categorized according to a PSI.** PSI 1 in non-AKI patients were significantly higher (*p<0.001) than all patients who developed AKI, regardless of the stage. Furthermore, in non-AKI patients PSI 3 and 4 were significantly († p<0.001) less than in any stage of AKI patients.

Besides, deaths in patients classified with AKI staging were significantly correlated with higher staging, as death percentages were 19.5%, 73%, and 86%, in stages 1, II, and III, respectively (p<0.001) (Fig 3). Furthermore, deaths in patients with AKI Stage I was significantly higher than patients with no AKI (p<0.001, OR 10.9 and 95% CI 3.5–34.5). Furthermore, death was correlated to severity of hypoalbuminemia and AKI staging (Fig 1C). The severity of hypoalbuminemia during AKI development was categorized as >35 g/L normal, vs. 32-<35 g/L, and <32 g/L. Multivariate regression analysis revealed that moderate to severe hypoalbuminemia (<32 g/L) was independently correlated to death in AKI patients with an OR of 10.99 (CI 95% 4.1–29.3, p<0.001). Besides, it has to be mentioned that death was not associated with a history of CKD (p>0.5, OR 0.81). Furthermore, 44 patients out of 64 who developed AKI stage III required renal replacement therapy (RRT), and 33 of the 44 patients died.

In the 87 deaths that occurred in COVID-19 patients with AKI, 68 patients (78.2%) were on mechanical ventilation (MV), and only 14 patients (17%) who were on MV survived. AKI patients who went through MV had a death OR of 17.24 (95% CI 7.94–37.04, p<0.001). In comparison, 8 non-AKI patients (4.3%) went through MV, and 5 (62.5%) patients survived, thus making death OR 111 in non-AKI patients (95% CI 9.25–1000, p<0.001).

The multivariate analysis in AKI patients showed that death is significantly related to the AKI stage (OR 8.37; 95% CI 2.96–23.64) and MV (OR 11.73; 95% CI 3.84–35.85) (p<0.0001), but not to PSI or RRT.

## ACEI/ARBS use protected from AKI II/III development

ACEI/ARBS treatment in hypertensive patients did not increase the PSI. Furthermore, data showed that overall patients, hypertensive or diabetic, who were taking ACEI/ARBS had less chance to develop AKI stage II/III with OR of 0.19–0.27 (p<0.05–0.01) (Table 6). In addition, patients who were taking ACEI/ARBS had less percent of mortality (25.8%) than patients who were not on ACEI/ARBS (58.8%) (OR = 0.244, CI 95% of 0.097–0.612, p< 0.01). Besides, patients who were under dexamethasone (except diabetic patients) or furosemide were

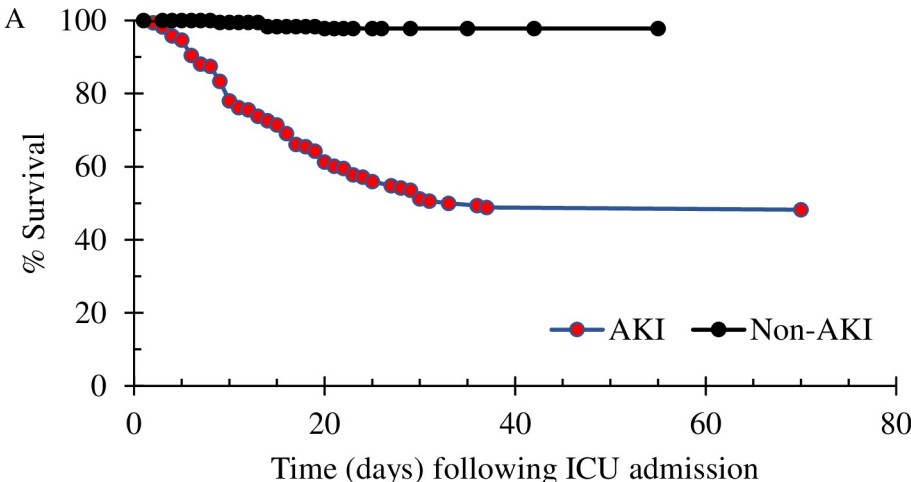

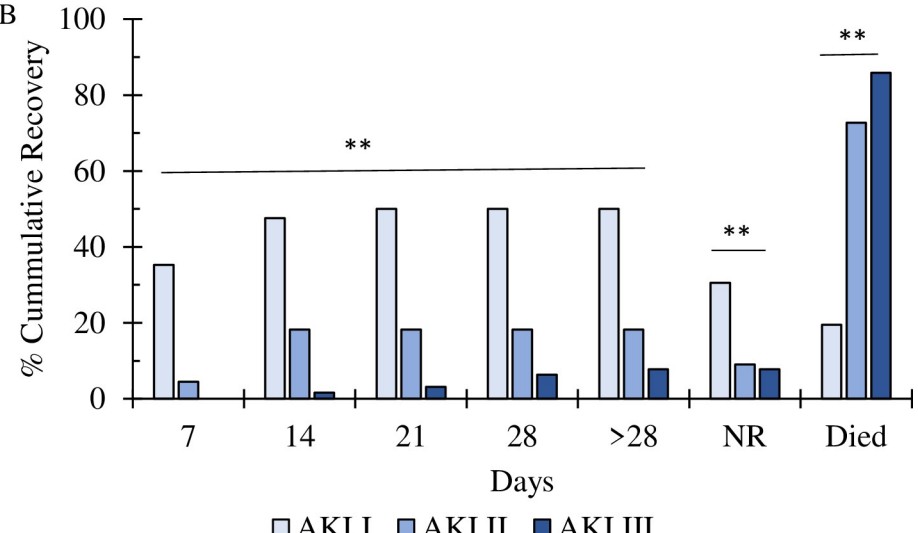

**Fig 3.** A. Kaplan-Meier plot of COVID-19 patients who did not develop AKI and those who did develop AKI. B. The percentage of cumulative recovery rate in COVID-19 patients with AKI is directly related to AKI staging. 35% of AKI stage 1 patients started to recover after 7 days of admission; this percentage went up to ~50% after 2 weeks and plateaued after that (**p<0.001 vs. stages 2 and 3). Besides, the percentages of patients who did not recover were significantly higher in stages I than stages II and III (** p<0.001). Lastly, deaths in patients classified with AKI staging were significantly correlated with higher staging (**p<0.001).

associated with AKI development stage I and stage II/III (Table 6). On the other hand, none of the other medications, such as lopinavir/ritonavir, interferon-β, ribavirin, enoxaparin, furosemide, and tocilizumab showed any association or protection with or against AKI development to staging I or II/III.

## Discussion

In this retrospective observational cohort study, 47.6% of COVID patients developed AKI, and 51.8% of AKI patients (vs. 2.2% in the non-AKI group) succumbed to the disease. This high mortality was associated with the AKI stage, whereby the death was highest among AKI stage III. These high mortality rates in AKI patients with COVID-19 were similar to other studies in

**Table 6. Multivariate logistic regression analysis of selected treatments associated with AKI development following adjustment with age.**

| Patients | Treatment | OR | 95% CI | P value |
|---|---|---|---|---|
| *Overall Patients* | | | | |
| AKI Stage I | ACEI/ARBS | 1.31 | 0.62–2.78 | n.s. |
| | Furosemide | 2.83 | 1.48–5.41 | <0.01 |
| | Dexamethasone | 3.06 | 1.48–6.33 | <0.01 |
| AKI Stage II+III | ACEI/ARBS | 0.27 | 0.10–0.73 | <0.01 |
| | Furosemide | 4.83 | 2.51–9.26 | <0.001 |
| | Dexamethasone | 2.76 | 1.32–5.79 | <0.01 |
| Non-AKI | | Ref | Ref | Ref |
| *Hypertensive* | | | | |
| AKI Stage I | ACEI/ARBS | 0.73 | 0.25–2.14 | n.s. |
| | Furosemide | 2.83 | 1.48–5.41 | <0.01 |
| | Dexamethasone | 4.56 | 1.25–16.52 | <0.01 |
| AKI Stage II+III | ACEI/ARBS | 0.19 | 0.05–0.19 | <0.05 |
| | Furosemide | 7.98 | 2.58–24.73 | <0.001 |
| | Dexamethasone | 2.76 | 1.32–5.79 | <0.05 |
| Non-AKI | | Ref | Ref | Ref |
| *Diabetic* | | | | |
| AKI Stage I | ACEI/ARBS | 0.81 | 0.29–2.28 | n.s. |
| | Furosemide | 4.06 | 1.55–10.63 | <0.01 |
| | Dexamethasone | 2.09 | 0.71–6.14 | n.s. |
| AKI Stage II+III | ACEI/ARBS | 0.19 | 0.05–0.19 | <0.01 |
| | Furosemide | 7.49 | 2.71–20.69 | <0.001 |
| | Dexamethasone | 2.24 | 0.72–6.95 | n.s. |
| Non-AKI | | Ref | Ref | Ref |

New York, USA, and Wuhan, China [5, 6, 8]. Furthermore, the present study showed a high incidence of AKI development in hospitalized patients, which is higher than the Hirsch et al. [5] study (AKI 36.6%) and by their completed study (AKI 39.8%) [6]. One primary reason for the difference is the small number of subjects in our study, 355 versus 5448 in the Hirsch study [5] and 9657 in their completed study [6]. On the contrary, a study by Cheng et al. [8] that was performed in Wuhan reported that AKI occurred only in 5.1% of the hospitalized patients. In the latter study, ethnicity could be a factor for the low AKI rate, however further studies in this direction are warranted.

The present study was conducted in one of the major hospitals in Bahrain. Bahrain is an Arab country and has a population of ~1.5 million, where immigrants make up approximately 48% of the total population [21]. In the present study, 82.1% of the AKI group were Bahrainis with an average age of 65.7 years old compared to 17.9% non-Bahrainis with an average age of 52.2 years. However, in multivariate logistic regression and with age adjustment, ethnicity was not associated with AKI development.

Similar to other studies, the present study showed that age, diabetes, hypertension, and having two or more comorbidities, including cardiovascular disease, increase the OR for developing AKI following SARS-CoV-2 infection [22]. Furthermore, several laboratory profiles upon admission were significantly different between those who developed AKI than those who did not. For instance, patients who developed AKI have high serum inflammatory, cardiac, or hypercoagulability markers and lymphopenia. One of the earlier studies (Feb 2020 -May 2020) on COVID-19 has shown similar laboratory results findings and related the laboratory results

to the prediction of clinical deterioration rather than AKI [23]. In the present study, however, and following age adjustment, only serum albumin, D-Dimer, Troponin I, and ProBNP high values were associated with AKI development. One of the recent studies showed that hypoalbuminemia predicts the outcome of COVID-19 independent of age and co-morbidity [24]. In that study, however, the authors did not study AKI as a reason for hypoalbuminemia but related that to inflammatory cascades and capillary permeability in severe cases of COVID-19 [24]. In contrast, other studies related hypoalbuminemia to nutritional index [25]. In our study, 21 patients (11%) who did not develop AKI, had hypoalbuminemia, and 2 of these patients died (p>0.05). On the contrary, death was correlated to the severity of hypoalbuminemia during AKI development (p<0.001). Since during AKI development, kidneys start excreting albumin in urine as seen herein, hypoalbuminemia becomes more evident and a predictor of severity and death of COVID-19.

Furthermore, in a meta-analysis study, elevated ProBNP on admission was associated with poor prognosis in COVID-19 patients [26]. Generally, ProBNP elevation indicates heart failure [27]. However, ProBNP elevation in COVID-19 patients may indicate myocardium tissue injury and inflammation, or its increase is due to hypoxia-induced pulmonary hypertension resulting an injury to the myocardium wall and therefore increasing the release of ProBNP. Besides, the association of high levels of ProBNP to AKI is related to low renal clearance [26]. However, our study showed that ProBNP levels were high on admission and before AKI development.

The present study showed that over 89–100% of AKI patients had a PSI scale of 2–5, compared to 88.5% of PSI scale 1–2, in non-AKI patients. These results suggest that AKI development, mainly AKI stages II and III in COVID-19 patients, are more related to pneumonia-induced right and/or left ventricular failures, which will lead to kidney congestion or kidney hypoperfusion, respectively [2, 9, 14]. Besides, in a post-mortem study of COVID-19 patients, SARS-CoV-2 RNA was found in 60% of COVID-19 patients, and the presence of RNA was associated with older age and accelerated deaths [28]. However, in the latter study, patients with SARS-CoV-2 negative RNA in their kidneys also died but had a longer survival time. These data suggest that SARS-CoV-2 renal tropism is not the only reason for AKI injury. Furthermore, SARS-CoV-2 infection-induced endothelialitis and hypercoagulability resulting in microthrombi or microembolism that cause kidney infarction is higher in AKI stage II and III [9, 16, 18]. On the other hand, since the rate of developing AKI following hospital admission was seen faster in AKI stage 1 patients than in AKI stage II or III [5], this may suggest a viremia-induced endothelial kidney injury mainly occurs in AKI stage I patients [13].

There have been concerns about the use of angiotensin ACEIs/ARBs in patients with COVID-19 that may increase the expression of ACE2, and therefore more infectivity of SARS-CoV-2 [29]. However, animal and human studies were inconclusive or did not show that the use of ACEIs/ARBs increases ACE2 expression [29]. On the other hand, our data suggested that using ACEIs/ARBs in hypertensive COVID-19 patients reduced AKI Stage II or III development. Furthermore, using ACEIs/ARBs was associated with low mortality and was not associated with pneumonia severity index. Although preliminary, these results studies should help to establish further studies on the use of ACEIs/ARBs in COVID-19 infected patients.

Since this study is observational, it is difficult to make inferences regarding causal relationships between exposure and AKI. Although the number of patients enrolled in this study is relatively good to the Bahrain population, the number of subjects enrolled is limited. Third, although the amount of validation, cleaning, auditing, and quality assurance was extensive during data collection, the data herein were extracted from electronic medical records using a data collection tabulation, including identifying AKI. Fourth, since all patients enrolled in this study were hospitalized, the generalization would be difficult.

## Conclusion

The prevalence of AKI in hospitalized COVID-19 patients in Bahrain and the mortality rate among AKI patients were high. In AKI patients, the mortality was higher in stages II and III. Furthermore, laboratory testing for serum albumin, hypercoagulability and cardiac injury markers maybe indicative of AKI development. Therefore, clinicians should be mandated to perform such tests on admission and follow-up in hospitalized patients.

## Supporting information

**S1 File. Patients' data.** COVID-19 Patients' data.
(XLSX)

**S2 File. Creatinine data.** Fig 1A and 1B creatinine data.
(XLSX)

**S3 File. Albumin data.** Fig 1C albumin data.
(XLSX)

**S4 File. PSI AKI and non-AKI data.** Fig 2 PSI AKI and non-AKI data.
(XLSX)

**S5 File. Survival data.** Fig 3A survival data.
(XLSX)

**S6 File. Recovery data.** Fig 3B recovery data.
(XLSX)

## Author Contributions

**Conceptualization:** Maryam N. Naser, Rana Al-Ghatam, Manaf M. Alqahtani.

**Data curation:** Hajar A. Alahmadi, Khalifa A. Mohamed.

**Formal analysis:** Maryam N. Naser.

**Methodology:** Maryam N. Naser.

**Supervision:** Nuria S. Perez.

**Validation:** Hajar A. Alahmadi, Khalifa A. Mohamed.

**Writing – original draft:** Maryam N. Naser, Rana Al-Ghatam, Manaf M. Alqahtani.

**Writing – review & editing:** Maryam N. Naser, Abdulla H. Darwish, Nahed K. Hasan, Nuria S. Perez.

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
