## [Decision Letter · Decision Letter 0]

14 Jul 2021

PONE-D-21-15637

Risk factors, predictions, and progression of acute kidney injury in hospitalized COVID-19 patients: An observational retrospective cohort study

PLOS ONE

Dear Dr. Naser,

Thank you for submitting your manuscript to PLOS ONE. After careful consideration, we feel that it has merit but does not fully meet PLOS ONE’s publication criteria as it currently stands. Therefore, we invite you to submit a revised version of the manuscript that addresses the points raised during the review process.

**The manuscript addresses a timely and potentially relevant topic. However, the study has several shortcomings that should be addressed to reach robust conclusions. To mention some of them, i) need to provide information with regard of prior immunosuppression; ii) need to include serum albumin in the data analysis; iii) need to quote and properly discuss the recent literature on the role of hypoalbuminemia as independent risk factor for COVID-19 AKI and severity; iv) need to perform further analysis including survival analysis for length of ICU stay, length of hospitalization and mortality; v) need to include and discuss the relevant papers on renal tropism of SARS-CoV-2 and AKI development.**

We look forward to receiving your revised manuscript.

Kind regards,

Giuseppe Remuzzi

Academic Editor

PLOS ONE

2. In your ethics statement in the Methods section and in the online submission form, please provide additional information about the data used in your retrospective study. Specifically, please ensure that you have discussed whether all data were fully anonymized before you accessed them and/or whether the IRB or ethics committee waived the requirement for informed consent. If patients provided informed written consent to have data from their medical records used in research, please include this information.

3. Thank you for providing the date(s) when patient medical information was initially recorded. Please also include the date(s) on which your research team accessed the databases/records to obtain the retrospective data used in your study.

4. "Please revise your tables to replace p-values of "n.s." to their actual numerical value.

7. Please amend your list of authors on the manuscript to ensure that each author is linked to an affiliation. Authors’ affiliations should reflect the institution where the work was done (if authors moved subsequently, you can also list the new affiliation stating “current affiliation:….” as necessary)

Reviewers' comments:

Reviewer's Responses to Questions

**Comments to the Author**

1. Is the manuscript technically sound, and do the data support the conclusions?

Reviewer #1: Yes

2. Has the statistical analysis been performed appropriately and rigorously? 

Reviewer #1: Yes

3. Have the authors made all data underlying the findings in their manuscript fully available?

Reviewer #1: Yes

4. Is the manuscript presented in an intelligible fashion and written in standard English?

Reviewer #1: Yes

5. Review Comments to the Author

Reviewer #1: In the present study, Naser et al. present data from a retrospective single-center cohort study identifying risk factors for AKI related to COVID-19. This is a timely and important topic in this pandemic. However, the following comments should be addressed to further improve the strength of this study:

1. Information with regard of prior immunosuppression should be included since this might also influence COVID-19 and AKI severity (e.g. B cell depletion).

2. In the data analysis, serum albumin should be included because hypalbuminemia has previously been described as independent risk factor for COVID-19 AKI and severity. In addition, the literature should be cited accordingly (Huang J et al. Hypoalbuminemiapredicts the outcome of COVID-19 independent of age and co-morbidity. J Med Virol. 2020, Aziz M et al. The association of lowserum albumin level with severe COVID-19: a systematic review andmeta-analysis. Crit Care. 2020, Paliogiannis P et al. Serumalbumin concentrations are associated with disease severity andoutcomes incoronavirus 19 disease (COVID-19): a systematic review and meta-analysis. Clin Exp Med. 2021, Tampe D et al. Urinary Levels of SARS-CoV-2 Nucleocapsid Protein Associate With Risk of AKI and COVID-19 Severity: A Single-Center Observational Study. Front Med. 2021).

3. Since AKI has been associated with COVID-19 severity and mortality, data analysis should also include survival analysis for ICU stay of length, length of hospitalization and mortality.

4. Since renal tropism of SARS-CoV-2 has been linked to AKI in COVID-19, the relevant literature should be discussed and included (Puelles et al. NEJM 2020, Braun et al. Lancet 2020).

6. PLOS authors have the option to publish the peer review history of their article (what does this mean?). If published, this will include your full peer review and any attached files.

Reviewer #1: No

---

## [Author Response · Author response to Decision Letter 0]

5 Aug 2021

The authors thank the reviewer(s) for addressing very important points. Therefore, in the revised version, we addressed all the points (highlighted in red). 

i) need to provide information with regard of prior immunosuppression; 

In the revised version (page 6), a statement is included stating “Furthermore, none of the patients had a history of immunodeficiency disorders.” 

ii) need to include serum albumin in the data analysis; 

In the revised version of the manuscript, a detailed analysis of serum albumin in AKI and non-AKI patients was included (Table 2, Table 3, Fig. 1C, Table 5) and also in page (2, 11, 12, 15).

iii) need to quote and properly discuss the recent literature on the role of hypoalbuminemia as independent risk factor for COVID-19 AKI and severity; 

In the revised version of the manuscript, a detailed discussion on the role of hypoalbuminemia is discussed (page 18). 

“In the present study, however, and following age adjustment, only serum albumin, D-Dimer, Troponin I, and ProBNP high values were associated with AKI development. One of the recent studies showed that hypoalbuminemia predicts the outcome of COVID-19 independent of age and co-morbidity [24]. In that study, however, the authors did not look into AKI as a reason for hypoalbuminemia but related that to inflammatory cascades and capillary permeability in severe cases of COVID-19 [24]. In contrast, other studies related hypoalbuminemia to nutritional index [25]. In our study, 21 patients (11%) who did not develop AKI, had hypoalbuminemia, and 2 of these patients died (p>0.05). On the contrary, death was correlated to the severity of hypoalbuminemia during AKI development (p<0.001). Since during AKI development, kidneys start excreting albumin in urine as seen herein, hypoalbuminemia becomes more evident and also a predictor of severity and death of COVID-19.”

iv) need to perform further analysis including survival analysis for length of ICU stay, length of hospitalization and mortality; 

Firstly, it has to be mentioned that all patients were admitted to the intensive care/isolation unit for COVID-19. Therefore, the length of hospitalization mentioned in the original manuscript refers to the length of stay in the ICU/isolation unit (see page 6 in the revised version).

Secondly, survival analysis with Kaplan Meier plot is introduced in the revised version showing % survival versus time (days) following admission (page 14 & Fig 3A).

“The Kaplan-Meier survival curve showed that 50% of patients with AKI died within 33 days of admission to the ICU/isolation unit (Fig 3a). It must be mentioned that there were 21 transferred patients, whom 6 out of 7 of them died (~7% of the death within AKI patients) within 2-28 days. This indicates that median survival time of 33 days is slightly underestimated. “ 

v) need to include and discuss the relevant papers on renal tropism of SARS-CoV-2 and AKI development.

In the revised version (page 19), we addressed this point. 

“Besides, in a post-mortem study of COVID-19 patients, SARS-CoV-2 RNA was found in 60% of COVID-19 patients, and the presence of RNA was associated with older age and accelerated deaths [28]. However, in the latter study, patients with SARS-CoV-2 negative RNA in their kidneys also died but had a longer survival time. These data suggest that SARS-CoV-2 renal tropism is not the only reason for AKI injury.”

---

## [Decision Letter · Decision Letter 1]

27 Aug 2021

Risk factors, predictions, and progression of acute kidney injury in hospitalized COVID-19 patients: An observational retrospective cohort study

PONE-D-21-15637R1

Dear Dr. Naser,

We’re pleased to inform you that your manuscript has been judged scientifically suitable for publication and will be formally accepted for publication once it meets all outstanding technical requirements.

**The revised manuscript has improved. The authors have properly addressed all the comments raised by the reviewer.**

Kind regards,

Giuseppe Remuzzi

Academic Editor

PLOS ONE

Additional Editor Comments (optional):

Reviewers' comments:

Reviewer's Responses to Questions

**Comments to the Author**

1. If the authors have adequately addressed your comments raised in a previous round of review and you feel that this manuscript is now acceptable for publication, you may indicate that here to bypass the “Comments to the Author” section, enter your conflict of interest statement in the “Confidential to Editor” section, and submit your "Accept" recommendation.

Reviewer #1: All comments have been addressed

2. Is the manuscript technically sound, and do the data support the conclusions?

Reviewer #1: Yes

3. Has the statistical analysis been performed appropriately and rigorously? 

Reviewer #1: Yes

4. Have the authors made all data underlying the findings in their manuscript fully available?

Reviewer #1: Yes

5. Is the manuscript presented in an intelligible fashion and written in standard English?

Reviewer #1: Yes

6. Review Comments to the Author

Reviewer #1: The authors adequately addressed all comments as suggested. Therefore, I recommend acceptance of the manuscript.

7. PLOS authors have the option to publish the peer review history of their article (what does this mean?). If published, this will include your full peer review and any attached files.

Reviewer #1: No

---

## [Editor Report · Acceptance letter]

20 Sep 2021

PONE-D-21-15637R1 

Risk factors, predictions, and progression of acute kidney injury in hospitalized COVID-19 patients: An observational retrospective cohort study 

Dear Dr. Naser:

I'm pleased to inform you that your manuscript has been deemed suitable for publication in PLOS ONE. Congratulations! Your manuscript is now with our production department. 

Kind regards, 

on behalf of

Prof. Giuseppe Remuzzi 

Academic Editor

PLOS ONE